# Characteristics and Crystal Structure of Calcareous Deposit Films Formed by Electrodeposition Process in Artificial and Natural Seawater

**Jun-Mu Park [1], Myeong-Hoon Lee [1] and Seung-Hyo Lee [2],***

1   Division of Marine Engineering, Korea Maritime and Ocean University, Busan 49112, Korea;
    qkrwnsan0305@gmail.com (J.-M.P.); leemh@kmou.ac.kr (M.-H.L.)
2   Department of Ocean Advanced Materials Convergence Engineering, Korea Maritime and Ocean University,
    Busan 49112, Korea
*   Correspondence: lsh@kmou.ac.kr

**Abstract:** In this study, we tried to form the calcareous deposit films by the electrodeposition process. The uniform and compact calcareous deposit films were formed by electrodeposition process and their crystal structure and characteristics were analyzed and evaluated using various surface analytical techniques. The mechanism of formation for the calcareous deposit films could be confirmed and the role of magnesium was verified by experiments in artificial and natural seawater solutions. The highest amount of the calcareous deposit film was obtained at $5 \, A/m^2$ while current densities between $1–3 \, A/m^2$ facilitated the formation of the most uniform and dense layers. In addition, the adhesion characteristics were found to be the best at $3 \, A/m^2$. The excellent characteristics of the calcareous deposit films were obtained when the dense film of brucite-$Mg(OH)_2$ and metastable aragonite-$CaCO_3$ was formed in the appropriate ratio.

**Keywords:** electrodeposition process; calcareous deposit films; aragonite crystal structure; seawater





## 1. Introduction

Corrosion damage and associated financial losses are increasing across the industry and a significant portion of cases are known to be due to marine corrosion. Especially, the economic loss from corrosion is estimated at $2.5 trillion worldwide, equivalent to 3.4% of world GDP. Therefore, if appropriate corrosion control and maintenance are possible, it is expected that 15% to 35% of the loss due to corrosion can be reduced [1]. The ports and offshore structures that mainly use metallic materials are used in harsh environments and continually suffer corrosion damage. Not only does corrosion have a deleterious influence on metallic structures, but it could also significantly impact the safety of human life and the pollution of natural environments. Therefore, in order to secure durability and corrosion resistance for long-term use of a marine structure, it is necessary to take appropriate measures for the structure based on the understanding of the marine corrosion environment [2,3].

Cathodic protection is the technique to control corrosion of the metal surface by making that surface the cathode in the electrochemical cell. Most of the process is progressed by the direct current impressed or the attachment of sacrificial anodes such as magnesium, aluminum, or zinc. It is also widely recognized as the most effective and technically appropriate corrosion prevention methodology for offshore structures and ships. For marine applications, steel must be protected against corrosion either by the application of the surface coating or by using cathodic protection when under fully immersed conditions, or the combination of both methods [4–9]. When applying cathodic protection in seawater, the compounds of $Mg(OH)_2$ and $CaCO_3$ are formed on the surface of the metal facilities. These mixed compounds are generally known as 'calcareous deposits'. This layer functions as a barrier against the corrosive environment, which leads to a decrease in the current

demand for cathodic protection [10–15]. Based on these advantages, partial application of calcareous deposit films to marine steel structures and concrete structures is under consideration. Additionally, some researchers have conducted various studies to apply the calcareous deposit films as the coating film and these results are detailed in [16–20]. However, the calcareous deposit films are partially formed on the surface of the cathode and there are some difficulties in maintaining both the corrosion resistance for a long period of time and the strong adhesion between deposits and the base metal. That is, the durability, precipitation rate, uniform adhesion, anti-corrosive effect, and the efficiency of calcareous deposit films have not yet been sufficiently proved [21–25]. For this reason, to apply calcareous deposit films, it is necessary to interpret the composition-crystal structure, adhesion property, and the long-term anti-corrosive effect of the calcareous deposit films under various environmental conditions. There are also several process factors that need to be supplemented in practical application design for field applications. In this study, we tried to form the calcareous deposit films by the electrodeposition process in artificial and natural seawater under various conditions. That is, the composition, crystal structure, and the morphology of the calcareous deposit films formed under various solutions and current density conditions were analyzed. In addition, the crystal structure, characteristics, and the formation mechanism of the calcareous deposit films are summarized from the viewpoint of the composition-crystal structure via appearance observation, weight gain, and adhesion evaluation.

## 2. Materials and Methods

In this experiment, a cold rolled steel sheet (SPCC, KS D 3512) was used. The steel sheet with a thickness of 0.1 mm was cut into 70 mm $\times$ 90 mm and then immersed in a 5% $H_2SO_4$ solution for 10 min to remove the mill scale and rust. The specimens were prepared by polishing with successively finer grades of abrasive paper down to 800 grades and washing with distilled water and acetone, and then dried. Additionally, a 1 mm-diameter hole was drilled at the top of each specimen to apply the cathode current: a copper wire that was connected and then insulated with silicon. The schematic diagram of the specimen used in the experiments is shown in Figure 1, and the chemical composition and mechanical properties of the specimens are shown in Table 1. In addition, artificial seawater was prepared by using ultrapure water (resistivity: about 6 M$\Omega$·cm) according to the ASTM D 1141-98 standard, which specifies the artificial seawater synthesizing method. The elements other than NaCl, $NaHCO_3$, $MgCl_2$, and $CaCl_2$ were not added for the purpose of precipitating the electrodeposits of the specific composition. Tables 2 and 3 show the compositional conditions of the artificial seawater used in the experiment and the average concentrations of the major elements in natural seawater. Current densities of 1, 3, and 5 A/m$^2$ for 24 h were applied to electrochemically prepare the coating layers, using a rectifier (ODP- 3032, OWON, Xiamen, China). In the case of the anode, the carbon rod, which is an insoluble anode, was used to prevent other components other than ions from dissolving in seawater from reduction. A solution comprising 1000 mL of artificial seawater (Ca-free, Mg-free, and Ca + Mg) and 1000 mL natural seawater was used. The calcareous deposits formed by each condition according to solution and current density were analyzed for composition elements, surface morphology, and the crystal structure. The calcareous deposit films were observed at 500, 1000, and 5000 magnification using the scanning electron microscope (FE-SEM, MIRA 3, Tescan, Brno, Czech Republic). At this time, the acceleration voltage of 5 to 15 kV was applied. In addition, for component analysis of the formed film, the energy dispersive X-ray spectroscopy (EDS, Octane Elect EDS, Mahwah, NJ, USA) was used to quantitatively analyze the calcareous deposit films in automatic mode at 500 to 1000 magnification. The crystal structure analysis of the formed calcareous deposit films was performed using the X-ray diffraction (XRD, Smartlap, Rikaku, Tokyo, Japan). X-rays were measured with Cu k$\alpha$ and the voltage and current of the X-ray tube were set to 40 kV and 40 mA, respectively. In addition, K-$\beta$ was used as the filter, and the scanning speed was 0.02 $^\circ$/min. At this time, the range of the measured 2$\theta$ value

was carried out from 20° to 80°. The measured results were compared with the JCPDS card to qualitatively analyze the film formed under each condition. The adhesion test was conducted using ISO standard tape and a knife and the durability of the calcareous deposits was evaluated by dividing the degree of deterioration and delamination of the film into 0 to 5 grades.

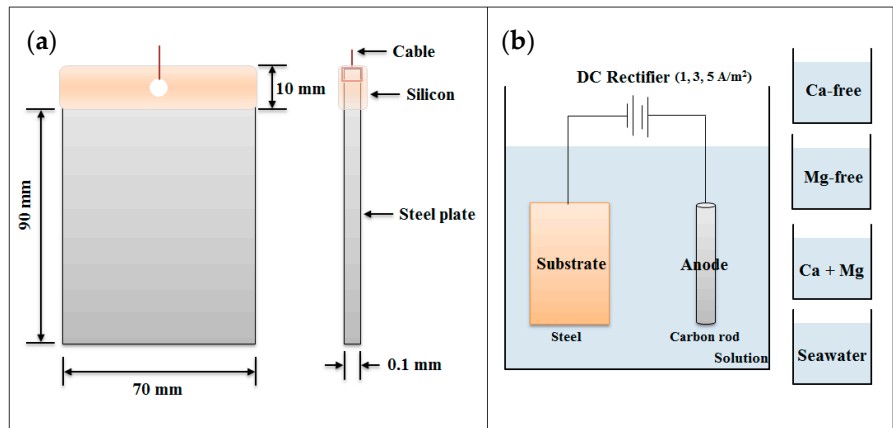

**Figure 1.** Experimental schematic diagram of electrodeposition process in artificial seawater and natural seawater: (**a**) schematic diagram of a specimen; (**b**) schematic diagram of electrodeposition experiment in various synthesized solutions.

**Table 1.** Chemical composition and mechanical properties of specimen.

| Materials (Steel Plate) | | SPCC (KS D 3512) | |
|---|---|---|---|
| **Chemical Composition (wt.%)** | | **Mechanical Properties** | |
| Fe | bal. | Yield Strength $(N/mm^2)$ | ~280 |
| C | 0.15 | | |
| Mn | 0.60 | Tensile strength $(N/mm^2)$ | ~410 |
| P | 0.05 | | |
| S | 0.05 | Elongation (%) | 28~ |

**Table 2.** Average chemical composition of the used natural seawater.

| Species | Concentration | | Species | Concentration | |
|---|---|---|---|---|---|
| | **mol/L** | **g/kg** | | **mol/L** | **g/kg** |
| $Na^+$ | 0.4685 | 10.77 | $Br^-$ | 0.000842 | 0.0673 |
| $K^+$ | 0.01021 | 0.399 | $F^-$ | 0.00068 | 0.0013 |
| $Mg^{2+}$ | 0.05308 | 1.29 | $HCO_3^-$ | 0.0023 | 0.14 |
| $Ca^{2+}$ | 0.01028 | 0.4121 | $SO_4^{2-}$ | 0.02823 | 2.712 |
| $Sr^{2+}$ | 0.00009 | 0.079 | $B(OH)_3$ | 0.000416 | 0.0257 |
| $Cl^-$ | 0.5459 | 19.354 | – | – | – |

**Table 3.** Chemical composition of the used artificial seawater.

| Ca-Free Solution | | Mg-Free Solution | | Ca + Mg Solution | |
|---|---|---|---|---|---|
| **Compound** | **Concentration (mol/L)** | **Compound** | **Concentration (mol/L)** | **Compound** | **Concentration (mol/L)** |
| NaCl | 0.411 | NaCl | 0.411 | NaCl | 0.411 |
| $MgCl_2$ | 0.0528 | $MgCl_2$ | – | $MgCl_2$ | 0.0528 |
| $CaCl_2$ | – | $CaCl_2$ | 0.0103 | $CaCl_2$ | 0.0103 |
| $NaHCO_3$ | 0.00186 | $NaHCO_3$ | 0.00186 | $NaHCO_3$ | 0.00186 |

## 3. Results

### 3.1. Appearance and Weight Gain of the Calcareous Deposit Films

Figures 2 and 3 show the results of the analysis of the appearance and weight gain for the calcareous deposit films formed by applying the cathode current through the DC rectifier for 24 h in artificial and natural seawater. As the applied current was increased to 1, 3, and 5 $A/m^2$, the amount of calcareous deposit films formed increased proportionally. However, even though the deposit mass was the highest at 5 $A/m^2$, delamination was also noticed, probably due to the strong hydrogen evolution process as a secondary cathodic reaction. That is, it was confirmed that when a relatively small amount of current density was applied, the calcareous film was uniformly and densely formed.

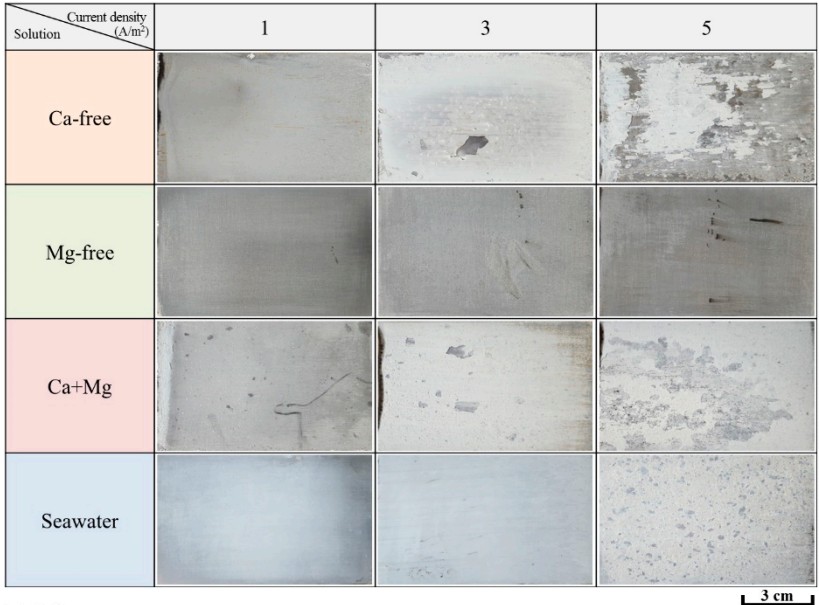

**Figure 2.** Photographs of calcareous deposit films according to the current density and solution.

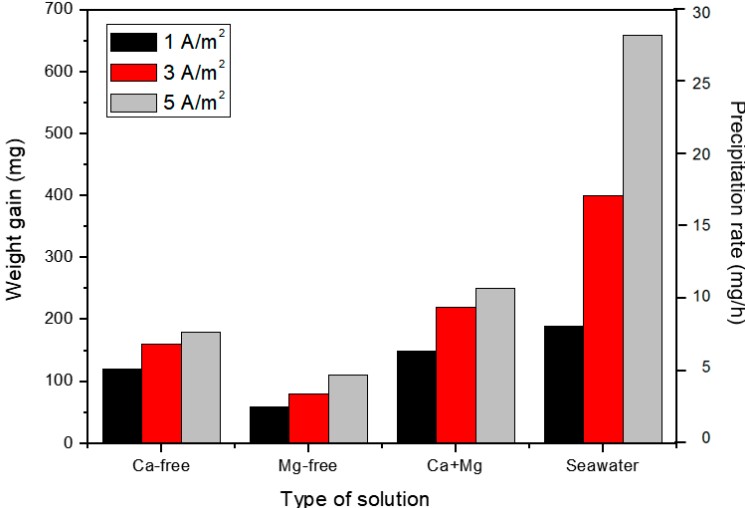

**Figure 3.** Variation of weight gain on calcareous deposit films in artificial seawater and natural seawater after 24 h.

According to the type of solution, the largest weight gain was observed in natural seawater. In the case of the artificial seawater, the Ca + Mg solution showed the highest weight gain because it was prepared with a composition like natural seawater. The result

showed that the amount of calcareous deposit films tended to increase in descending order of natural seawater, Ca + Mg solution, Ca-free solution, and Mg-free solution. Notably, it was confirmed that the weight gain of the Mg-free solution was significantly smaller than that of other solutions. This is probably because the formation of the $CaCO_3$ nucleus-films is not easy due to the absence of the $Mg(OH)_2$ film.

### 3.2. Composition, Crystal Structure, and Morphology of the Calcareous Deposits

Figure 4 shows the EDS composition analysis results of the electrodeposited films formed in artificial and natural seawater at different current density values. All the investigated deposits contain Mg, Ca, O, Na, and Cl as major elements. Notably, it was confirmed that the amounts of Mg and Ca components to be detected vary depending on the level of the applied current. That is, the Mg component was mainly detected when a high cathode current was applied and an increased amount of the Ca component was confirmed when a relatively low cathode current was applied. Table 4 shows the data of the composition ratio of Ca to Mg. It is confirmed that the ratio of Ca to Mg increases when the relatively low cathode current is applied. That is, it is considered that when the low cathode current is applied, not only the relatively thin film but also the film having a high Ca content is formed. In addition, it was confirmed in the results of the analysis for the relative intensity ratios of aragonite-$CaCO_3$/brucite-$Mg(OH)_2$ in the Ca + Mg solution and natural seawater. As with the previous composition ratio, the relative intensity ratio was increased when a relatively low cathode current was applied. That is, the ratio was the highest under the condition of 3 $A/m^2$. Although it is a relatively thin film, a high aragonite-$CaCO_3$ ratio is believed to form.

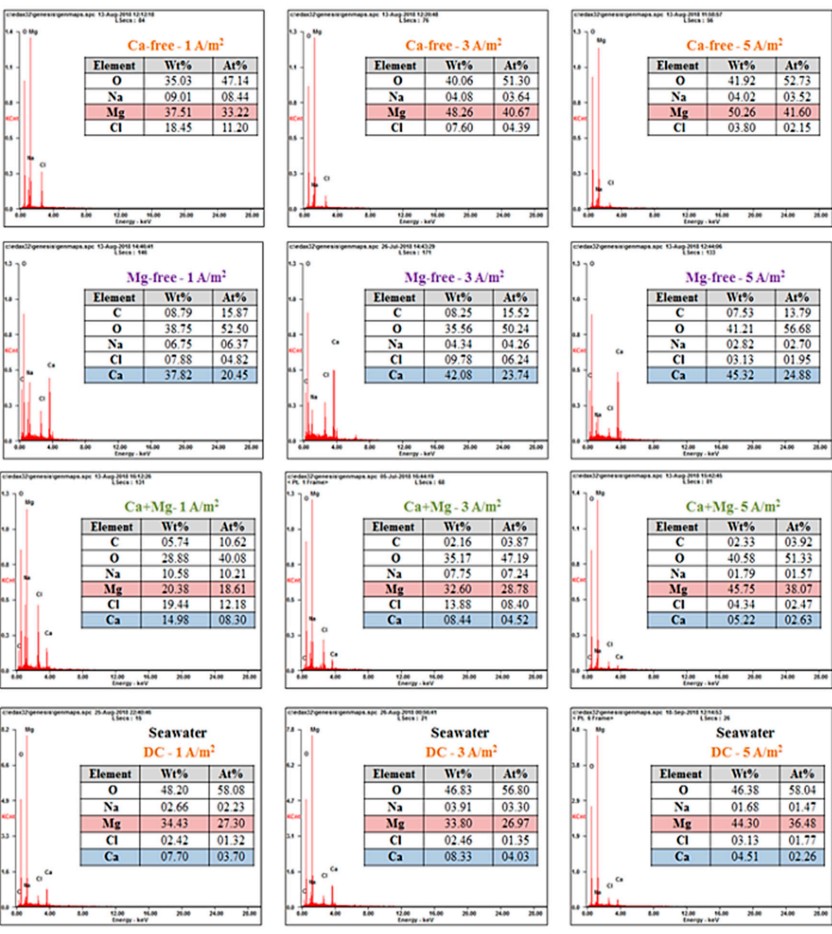

**Figure 4.** Elemental composition analysis of calcareous deposit films according to the current density for 24 h.

**Table 4.** Composition ratio (Ca/Mg) analysis of calcareous deposit films after 24 h.

| Current Density | Composition Ratio of Ca/Mg | | Composition Ratio of Aragonite-CaCO$_3$/Brucite-Mg(OH)$_2$ | |
|---|---|---|---|---|
| | Ca + Mg | Seawater | Ca + Mg | Seawater |
| 1 A/m$^2$ | 0.44 | 0.22 | 0.227 | 0.270 |
| 3 A/m$^2$ | 0.26 | 0.25 | 0.279 | 0.405 |
| 5 A/m$^2$ | 0.11 | 0.10 | 0.153 | 0.103 |

The results of the XRD crystal structure analysis are shown in Figure 5. It is confirmed that the brucite crystal structure of Mg(OH)$_2$ and the calcite and aragonite crystal structure of CaCO$_3$ via the crystal structure analysis of the calcareous deposit films formed in artificial and natural seawater. In the Ca-free solution, the brucite crystal structure of Mg(OH)$_2$ was detected in almost all cases and it is considered that brucite crystals are uniformly distributed through analysis of the diffraction peak intensity. CaCO$_3$ was mainly precipitated in the Mg-free solution. While the CaCO$_3$ detected was mainly grown as the calcite, aragonite growth did not proceed. However, it was confirmed that CaCO$_3$ is present as an aragonite crystal structure rather than calcite in the Ca + Mg solution and natural seawater in which both Ca$^{2+}$ and Mg$^{2+}$ are present. From these results, it can be conjectured that Mg$^{2+}$ affects the crystal structure of CaCO$_3$. That is, aragonite-CaCO$_3$ of metastable is formed due to the absorption and occlusion of Mg$^{2+}$ in the Ca crystal lattice. It is considered that Mg$^{2+}$ inhibits the formation and growth of the calcite crystal nucleus while promoting aragonite crystal growth [26–28].

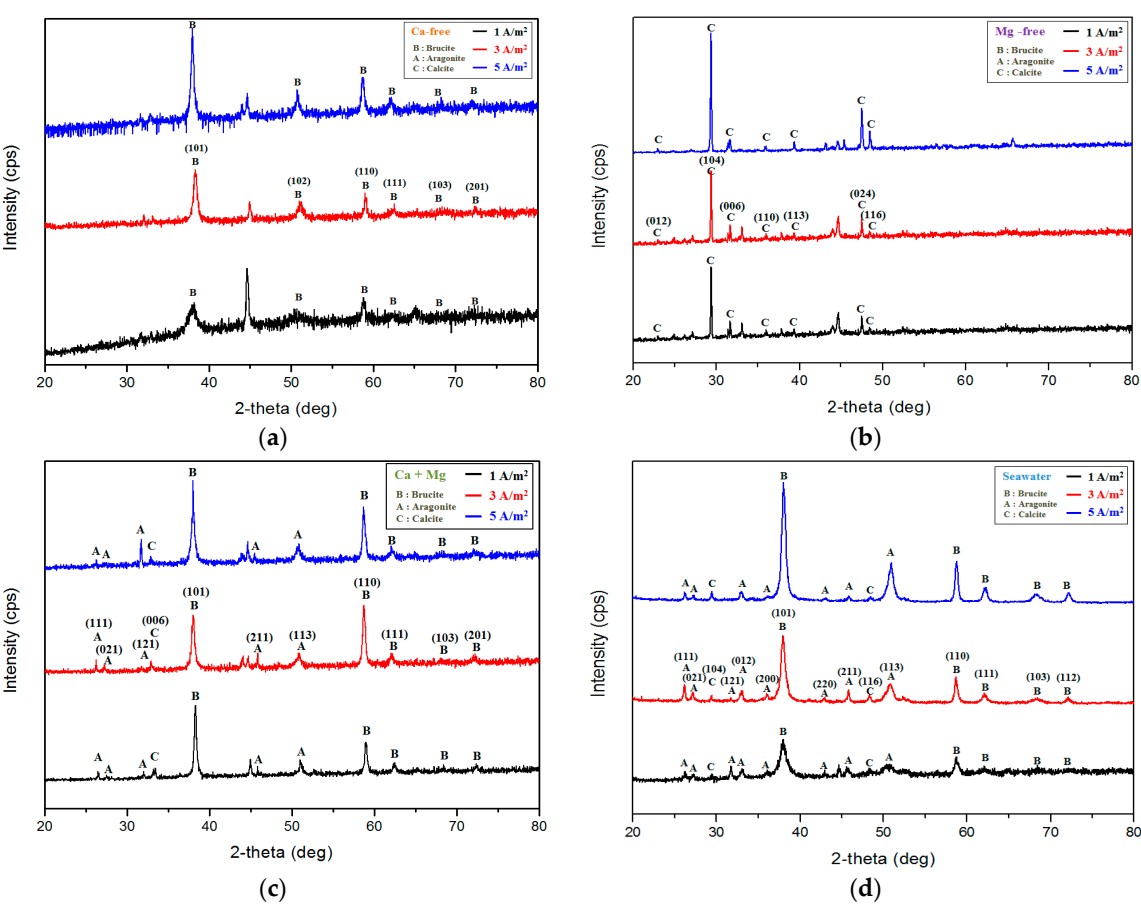

**Figure 5.** Crystal structure of the calcareous deposit films after 24 h in each solution (**a**) Ca-free, (**b**) Mg-free, (**c**) Ca + Mg, (**d**) natural seawater.

Figure 6 shows the results of surface morphology analysis for calcareous deposit films formed in artificial and natural seawater. In the Ca-free solution, $Mg(OH)_2$ of the brucite crystal structure was formed, and in the Mg-free solution, $CaCO_3$ was synthesized to the calcite crystal structure. In addition, in the Ca + Mg solution and natural seawater, it was confirmed that $CaCO_3$ of aragonite crystal structure was formed on $Mg(OH)_2$ film by the influence of $Mg^{2+}$. As a result, it is thought that $Mg^{2+}$ is combined by adsorption and occlusion on the high surface energy plane, which inhibits the growth of calcite crystals. Increasing the applied current tended to inhibit crystal growth and change in the crystal structure due to the hydrogen evolution reaction at the cathode. In addition, partially non-uniform areas were observed.

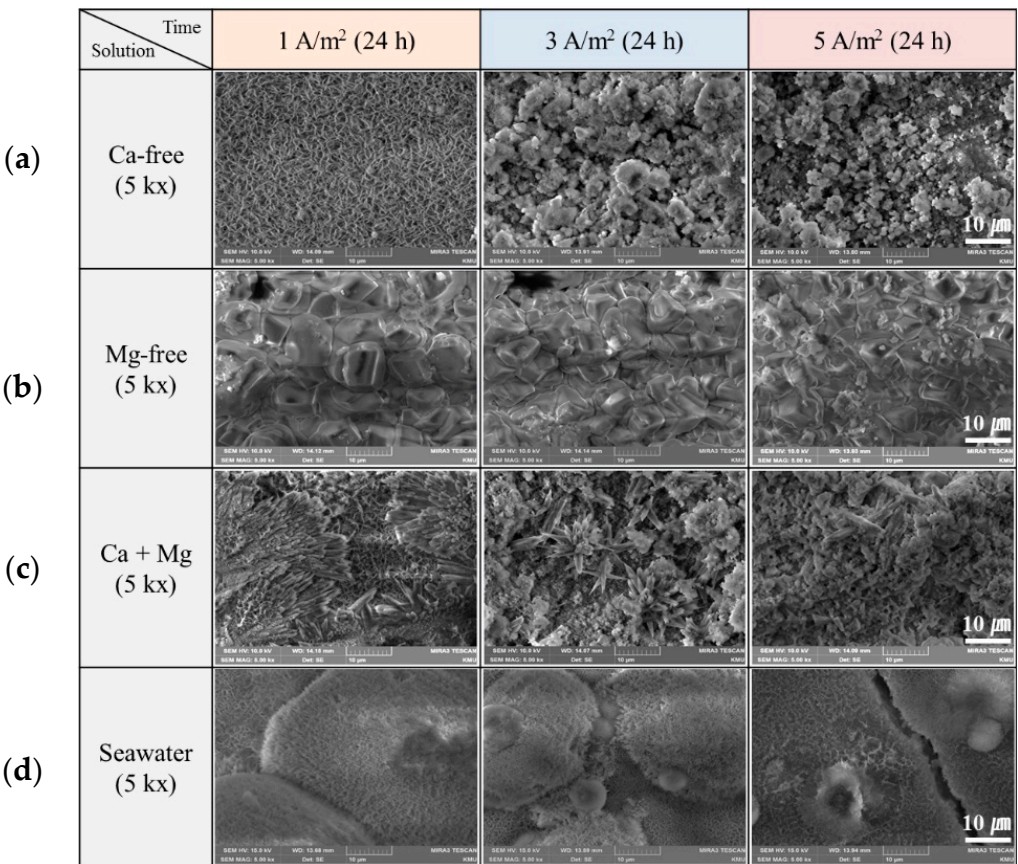

**Figure 6.** Morphology of the calcareous deposit films after 24 h in each solution (**a**) Ca-free, (**b**) Mg-free, (**c**) Ca + Mg, (**d**) natural seawater.

### 3.3. Evaluation of Adhesion Properties of Calcareous Deposit Films

The adhesion test results of the calcareous deposit films formed in artificial and natural seawater under experimental conditions are shown in Figure 7. The adhesion test was carried out with the taping test according to the ISO 2409 standard.

The adhesion properties of brucite-$Mg(OH)_2$ in the Ca-free solution and calcite-$CaCO_3$ in the Mg-free solution were not good compared to the other solution. In addition, the thickest film was formed when the relatively high cathode current was applied. However, the calcareous deposit films could not be stably bonded to the substrate and partial delamination occurred due to the active hydrogen-evolution reaction. That is, when the brucite-$Mg(OH)_2$ film was formed, it showed weak adhesion characteristics and it was confirmed that the adhesion characteristics were changed depending on the calcite or aragonite crystal structure.

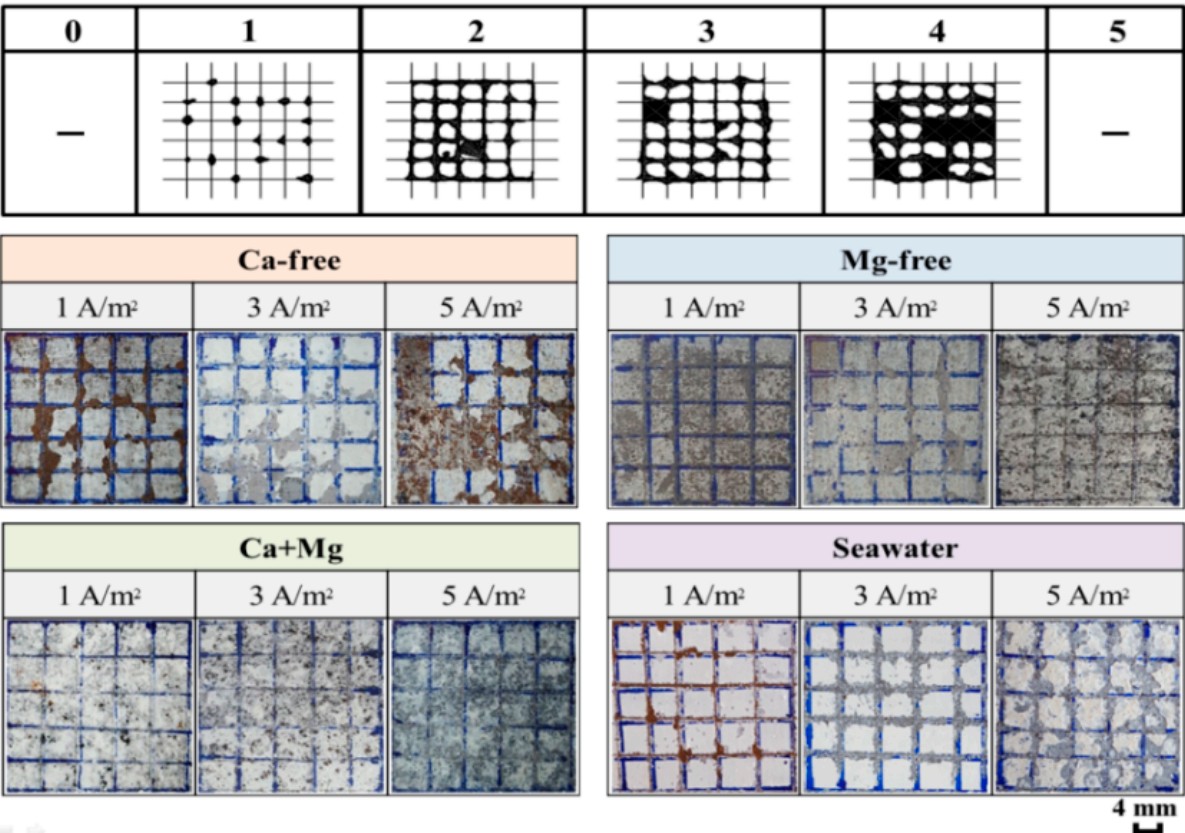

**Figure 7.** Taping test (ISO 2409) for adhesion properties of calcareous deposit films after 24 h.

The weight gain of the calcareous deposit films formed by Ca + Mg solution and natural seawater increased as the cathode current increased but the adhesion properties were best at 3 A/m$^2$. However, in terms of the composition and crystal structure, the ratio of Ca/Mg and the relative strength of aragonite-CaCO$_3$/brucite-Mg(OH)$_2$ are high. That is, brucite-Mg(OH)$_2$, which is the plate-like relatively soft film, and aragonite-CaCO$_3$ are formed into a tightly bonded film. Therefore, it is considered that adhesiveness is improved.

## 4. Discussion

The formation mechanism of the calcareous deposit films via the composition, crystal structure, and the morphology analysis-evaluation are summarized in Figure 8. When the cathodic current is applied, the reduction reaction of oxygen and water occurs on the surface of the cathode. As a result, in the diffusion layer between the metal surface and the solution interface, relatively high OH$^-$ is generated, producing an increase of the pH. The OH$^-$ generated in this way reacts with Mg$^{2+}$ to form the Mg(OH)$_2$ film preferentially. Due to the formed film, the amount of OH$^-$ is relatively reduced on the surface of the substrate. Therefore, the pH is lowered locally to form the CaCO$_3$ film. At this time, when the cathode current is high, the large number of generated OH$^-$ reacts with Mg$^{2+}$, thereby continuously forming the Mg(OH)$_2$ film, increasing the amount of brucite-Mg(OH)$_2$ on the surface. On the other hand, in the case of a low cathodic current, the reduced electrons on the surface of the substrate induce the binding of CO$_3$$^{2-}$ with Ca$^{2+}$ in the solution and the amount of argonite-CaCO$_3$ on the surface is considered to be increased [29–33].

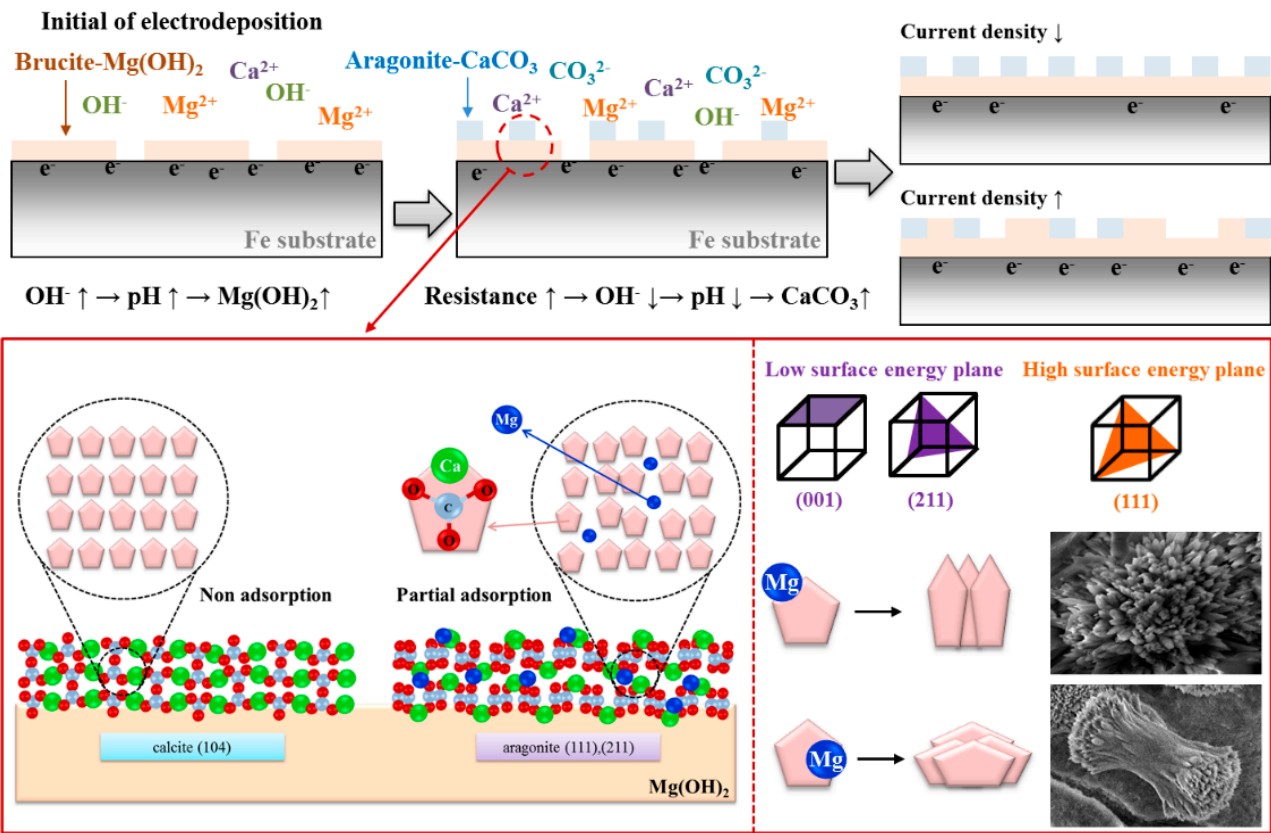

**Figure 8.** Formation mechanism of calcareous deposit films according to the composition and crystal structure.

Based on the above results, the surface energy plane and surface energy values of the aragonite crystal structure are shown in Figure 9 to investigate the cause of the CaCO$_3$ crystal structure change of the calcareous deposit films that are formed. The representative surfaces with low surface energy are (001) and (211) planes and the representative surfaces with high surface energy are (111) planes [34,35]. The standard plane spacing d-value on the JCPDS card was compared with the plane spacing d-value on the high and low surface energy of the calcareous deposit films. The comparison was conducted to confirm the solid solubility, which is the inhibition effect, by the adsorption and occlusion of Mg$^{2+}$ on the CaCO$_3$ crystal structure. Figure 9 shows the mechanism of the crystal structure change for the CaCO$_3$ film due to the inhibition effect caused by Mg$^{2+}$. It was found that the inter-atomic distances decreased with increasing Mg$^{2+}$ concentrations in both the high surface energy plane (111) and the low surface energy plane (211). The above results suggest that Mg$^{2+}$ is combined by adsorption and occlusion on the high surface energy plane, which inhibits crystal growth. As a result, it is considered that the crystal growth of CaCO$_3$ film occurs around the surface with a relatively low surface energy plane.

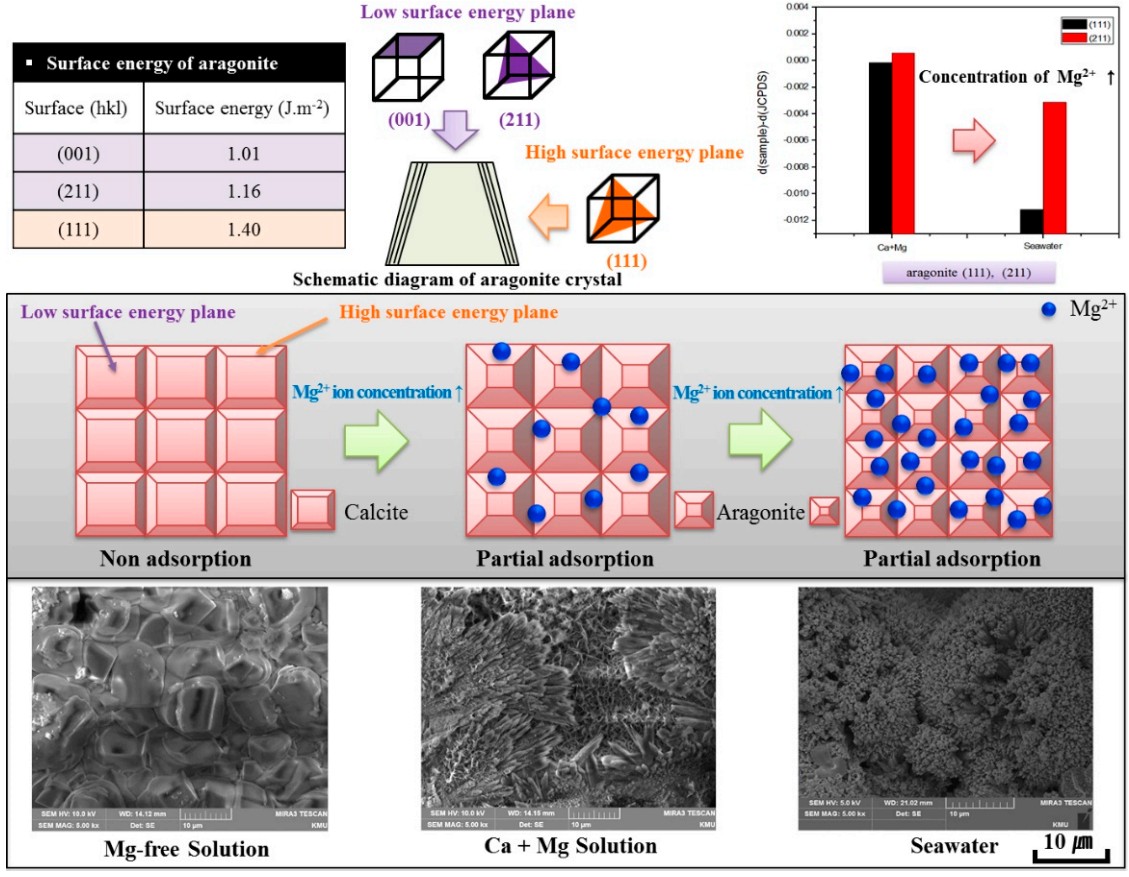

**Figure 9.** Schematic diagram of mechanism for changing of crystal structure in CaCO$_3$ films by inhibition effect.

## 5. Conclusions

The crystal structure and characteristics of the calcareous deposit films formed by electrodeposition process in artificial and natural seawater were studied, and the obtained results are summarized as follows.

- When the cathodic current is applied, the Mg(OH)$_2$ film is preferentially formed in the initial stage and thereafter, aragonite-CaCO$_3$ is formed due to the locally decreased pH. That is, the electrons on the surface of the substrate form OH$^-$ by the reduction reaction of oxygen and water in the solution. The generated OH$^-$ is preferentially formed into the Mg(OH)$_2$ film in combination with Mg$^{2+}$. When the film is formed, the pH locally decreases because of the relatively reduced OH$^-$ on the substrate surface. Therefore, the reaction of CaCO$_3$ formation could dominate over the reaction of Mg(OH)$_2$ formation at this time.

- It was confirmed that the crystal structure of the calcareous deposit films formed in the Ca + Mg solution and natural seawater increased the formation of aragonite more than calcite. It is considered that Mg$^{2+}$ inhibits the formation and growth of the calcite nucleus while promoting aragonite crystal growth. That is, aragonite-CaCO$_3$ of metastable is formed due to the adsorption and occlusion of Mg$^{2+}$ in the Ca crystal lattice.

- In the electrodeposition process, the precipitation amount of the formed film increased as the cathode current was larger; however, the adhesion characteristics were found to be the best at 3 A/m$^2$. That is, the excellent characteristics of the calcareous deposit films were obtained when the dense film of brucite-Mg(OH)$_2$ and metastable aragonite-CaCO$_3$ was formed in the appropriate ratio.

- The optimal conditions derived from the research results provide more practical design guidelines that can improve the adhesion characteristics of calcareous deposit

films. If the calcareous deposit films with the increased aragonite-$CaCO_3$ ratio are applied, the calcareous deposit films are expected to extend the lifetime of sacrificial anodes and reduce maintenance costs. In addition, it is expected that the application scope of cathodic protection such as ships and offshore structures can be expanded.

**Author Contributions:** Conceptualization, J.-M.P. and M.-H.L.; methodology, M.-H.L.; software, J.-M.P.; validation, J.-M.P., M.-H.L. and S.-H.L.; formal analysis, J.-M.P.; investigation, J.-M.P.; resources, J.-M.P.; data curation, J.-M.P.; writing—original draft preparation, J.-M.P.; writing—review and editing, S.-H.L.; visualization, S.-H.L.; supervision, S.-H.L.; project administration, S.-H.L.; funding acquisition, S.-H.L. All authors have read and agreed to the published version of the manuscript.

**Funding:** This research received no external funding.

**Institutional Review Board Statement:** Not applicable.

**Informed Consent Statement:** Not applicable.

**Data Availability Statement:** Data is contained within the article.

**Acknowledgments:** This research was a part of the project titled 'The development of marine-waste disposal system optimized in an island-fishing village', funded by the Ministry of Oceans and Fisheries, Korea.

**Conflicts of Interest:** The authors declare no conflict of interest.

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
