# Peer review of "Characteristics and Crystal Structure of Calcareous Deposit Films Formed by Electrodeposition Process in Artificial and Natural Seawater"

_coatings, doi:10.3390/coatings11030359_

Round 1

Reviewer 1 Report

The manuscript concerns calcareous deposit films formed by electrodeposition process.

In general, the manuscript is interesting and well prepared. However, the introduction is relatively short, the cited literature is sparse (additionally, [5] and [6] are duplicated).

Below is a list of my comment:

  • Table 2; the title of the table does not match its content, additionally, the table is unreadable (large amount of data, different units).
  • Table 3 suggests that each solution was deposited at only one current density value.
  • References; Please correct duplicate items when formatting the references.
  • Superscripts and subscripts are missing throughout the manuscript (current density unit, notation of chemical formulas and ions). Please correct.

Author Response

Please see the attachment and Thank you so much. 

Reviewer 2 Report

This paper discusses the crystal structure of the calcareous deposit films formed by electrodeposition process in artificial and natural seawater. The paper is well organized, and the following are some detailed questions.

Is it possible to look at the SEM images of the cross-section of the calcareous deposit film in the seawater condition?  It may prove that the initial deposit is brucite, and the second layer is aragonite.

How could you approve this “the relative reduction of electrons on the substrate surface with the increase of the film resistance induce the bonding of CO32- with Ca2+ in the solution?” Why?

It is better to adjust the superscript of all the ions throughout the whole manuscript.

Author Response

(The authors gave the same response as above.)

Reviewer 3 Report

A considerable amount of papers has already been published regarding the formation of calcareous deposit during cathodic protection of maritime constructions as well as factors influencing this formation. The deposit is constituted by calcium carbonate and magnesium hydroxide, and it behaves like a barrier against corrosion in seawater. This work presents the results of calcareous deposit formation by cathodic protection of a cold rolled steel in artificial seawater (Ca-free, Mg-free, and Ca+Mg) and natural seawater by electrolysis with 1, 3, and 5 A/m2 for 24 hours. The crystal structure and characteristics (appearance, weight gain, and adhesion) were determined and discussed. The optimal conditions for the most dense and adhesive film of brucite-Mg(OH)2 and metastable aragonite-CaCO3 were found at 3 A/m2 current density. In general, the work is novelty and contains interesting results and interpretations.

However, several corrections and improvements are recommended to be made, as shown below:

1) Abstract: most of the text must be deleted, can enter the Introduction. The abstract must be redone by starting with: In this study, we tried to form the calcareous deposit films by the electrodeposition process… Then, it should continue with what results were obtained, in short.

2) Introduction: The cited papers are in general old publications about the subject. Therefore, some recent papers may be discussed in Introduction, examples:

-K. Zakowski, M. Szocinski, M. Narozny, Study of the formation of calcareous deposits on cathodically protected steel in Baltic Sea water, Anti-Corros. Methods Mater., 60 (2013)  95–99. 

-H. Karoui, B. Riffault, M. Jeannin, A. Kahoul, O. Gil, M. Ben Amor, M.M.Tlili, Electrochemical scaling of stainless steel in artificial seawater: role of experimental conditions on CaCO3 and Mg(OH)2 formation, Desalination 311 (2013) 234–240.

-C. Li, M. Du, J. Qiu, J. Zhang, Influence of temperature on the protectiveness and morphological characteristics of calcareous deposits polarized by galvanostatic mode, Acta Metall. Sin., 27 (2014) 131–139.

-Y. Yang, J.D. Scantlebury, E.V. Koroleva, A study of calcareous deposits on cathodically protected mild steel in artificial seawater, Metals, 5(1) (2015) 439-456.

-N. Ce, S. Paul, The effect of temperature and local pH on calcareous deposit formation in damaged thermal spray aluminum (TSA) coatings and its implication on corrosion mitigation of offshore steel structures, Coatings, 7(4) (2017) 52.

-D.N. Dang, S. Gascoin, A. Zanibellato, C.G. Da Silva, M. Lemoine, B. Riffault, R. Sabot, M. Jeannin, D. Chateigner, O. Gil, Role of brucite dissolution in calcium carbonate precipitation from artificial and natural seawaters, Cryst. Growth Des., 17(4) (2017) 1502–1513.

-C.A. Loto, Calcareous deposits and effects on steels surfaces in seawater – (A review and experimental study), Oriental Journal of Chemistry, 34(5) (2018) 2332-2341.

-C. Carre, A. Zanibellato, M. Jeannin, R. Sabot, P. Gunken-Grillon, A. Serres, Electrochemical calcareous deposition in seawater. A review, Environmental Chemistry Letters, 18(4) (2020) 1193-1208.

3) Materials and Methods:

-Lines 80, 81: write mL instead of ml

-Table 2: the units of concentrations of ions (Na+, K+, etc) are strange; please correct, especially

mmol-1kg-1

-Delete Table 3; it is useless, it does not bring new information

-At the end of this section: information regarding equipment for weight gain, elemental analysis, XRD, microscopy, including adhesion procedure, must be written.

4) Results:

-Explain how the deposition rate of the calcareous deposit films was confirmed to be in good agreement with Faraday’s law? What are the calculations?

-L.108-112: Please check if it is an ascending order regarding the amount of calcareous deposit, because you state: …the weight gain of the Mg-free solution is significantly smaller than that of other solutions.

-Correct on the ordinate axis (in right) of Fig. 3: Precipitation rate, instead of ‘velocity’

-L.142: correct English language in the sentence, if more nuclei of calcite are formed: ..the formation and growth of the calcite nucleus, while promoting aragonite growth. Correct English language in many other sentences.

-Fig. 4 is very unclear and the caption (L.162) must explain the differences between the three columns

-Figs. 5 and 7 must be deleted, they contain only three experimental points; values of both composition ratios should be presented in a text or a new Table.

5) Discussion:

L.197-199: the sentence is trivial, considerations about transport of electrons and their activity on surface can be erased

L.203-205: the sentence is unclear: ..And the increased film resistance due to the formation of the Mg(OH)2 film results in relatively low OH- because of the reduced electrons on the substrate surface.

You should explain ‘increased film resistance’ (is an electrical resistance, or corrosion resistance??) and the ‘reduced electrons’ what does it mean??

-L.212-213: you should explain the sentence: ..Based on the above results, the surface energy plane and surface energy values of the aragonite crystal structure… You have calculated them for aragonite? (see Table inside Fig.11)

-Explain the differences: ‘d(sample) – d (JCPDS card)’ on the ordinate axis of Fig. 11, for Ca+Mg solution and for seawater

6) Conclusions:

-It is unclear the text and must correct:

L.239-240; L.242: ..Subsequently, the relative reduction of electrons on??

L.244: it is probably: ..Therefore, the CaCO3 nuclei are predominantly formed.

L.247-248: .. and growth of the calcite nucleus, while promoting aragonite growth.

L.254: .. and the relative strength of aragonite-CaCO3 /brucite-Mg(OH)2 are

7) References

-The authors' first names are written with initials, in: [3, 7, 8, 11, 12]

-The titles of cited papers are written in lowercase, in:  [3, 5, 11, 12]

-Reference [6] must be deleted, it is the repetition of [5]

-Other corrections:

1. D.F. Hasson, C.R. Crowe (Eds.), Materials for Marine System and Structures: Treatise on Materials Science and Technology, volume 28, Academic Press, 1998, p. 46.

3. H. Kobayashi, T. Yamaji, Y. Akira, H. Hamada, N. Mochizuki, K. Shimo, Study on retrofit design of port steel structures with cathodic protection of galvanic anode system, Corrosion Engineering, 62(2011) 192-197.

5. W.H. Hartt, C.H. Culberson, S.W. Smith, Calcareous deposits on metal surfaces in seawater—A critical review, Corrosion, 40(1984) 609–618.

Author Response

(The authors gave the same response as above.)

Reviewer 4 Report

In paper : Cathodic protection is the technique to control corrosion of the metal surface by making that surface the cathode in the electrochemical cell, the authors present some interesting experimental results. However I have few concerns before the publication : 

  • Major concern: The authors treat very badly the References section: Ref. 5 and 6 are the same, references 11 and 12 are missing from the text , appear only at the end, 12 references are a small number of references for this topic.
  • L159: erase phrase : All figures and tables .......etc.
  •  Minor concern:
  • L (line)77 , L94, L138, 139 etc and in many different places the authors must use subscript and superscript 
  •  L120 or L162/Figure 4: EDS experiments conditions must be specified : mode (automatic or element list), the detector error for each element, Standard deviation of the measurements 
  • L130: please re-phrase : It is confirmed ......
  •  L177-178: erase phrase Figure 9 .... seawater - is the same as the phrase from the last paragraph 

Author Response

(The authors gave the same response as above.)

Round 2

Reviewer 1 Report

I would like to thank the authors for considering my comments. The publication may be published in present form.  

Author Response

Thank you for your comments.

Reviewer 3 Report

The authors made the most of the recommended corrections.

However, some modifications are suggested as it follows.

The Abstract:

Please reformulate:

“In this study, we tried to form the calcareous deposit films by the electrodeposition process. The uniform and compact calcareous deposit films were formed by electrodeposition process and their crystal structure and characteristics were analyzed and evaluated using various surface analytical techniques. The mechanism of formation for the calcareous deposit films could be confirmed, and the role of magnesium was verified by experiments in artificial and natural seawater solutions. The highest amount of the calcareous deposit film was obtained at 5 A/m2, while current densities between 1-3 A/m2 facilitated the formation of the most uniform and dense layers. In addition, the best adhesion characteristics were evidenced for deposits electrodeposited at 3 A/m2. The excellent characteristics of the calcareous deposit films were obtained when the dense film of brucite-Mg(OH)2 and metastable aragonite-CaCO3 was formed in the appropriate ratio.”

Introduction:

Line 51: Please reformulate: “Also, some researchers have conducted various studies to apply the calcareous deposit films as the coating film and these results are detailed in [16-20].

Materials and Methods:

Lines 82-83: Please correct: Table 2 and 3 present the composition of the natural seawater, respectively of the artificial seawater that have been used as electrolytes during the electrodeposition process (Please provide the concentrations in mol/L for both tables to be homogeneous data)

Lines 83-84: Please reformulate: “Current densities of 1, 3, and 5 A/m2 for 24 hours have been applied to electrochemically prepare the coating layers, using a rectifier (please mention the producer of the rectifier)”

Please reformulate the caption of Table 2: Average chemical composition of the used natural seawater

Please reformulate the caption of Table 3: Chemical composition of the used artificial seawater

Lines 90-101: Please provide the details of the used equipment: FE-SEM, EDS, XRD

Please provide details related to the determination of the deposit mass involving gravimetric method.

Results:

Lines 119-121: Please reformulate:  “ Figure 2 presents the photographic images of the calcareous layers electrodeposited for 24 h in the involved electrolytes at different applied current densities. (Here you should detail some aspects related to the appearance of the deposits, such as uniformity, homogeneity, etc.). Figure 3 illustrates the dependence of the calcareous deposit mass against the used electrolyte for the applied current density values.

Please delete L123-125: “In general, the deposition rate of the calcareous deposit films was confirmed to be in good agreement with Faraday’s law, which shows a linear relationship with the current density when the time is constant.” This phrase doesn’t bring any significant information.

Lines 125-127: Please reformulate: “However, even the deposit mass was the highest at 5 A/m2, delamination was also noticed, probably due to the strong hydrogen evolution process as a secondary cathodic reaction.”

Lines 143-145: please correct: “Figure 4 shows the EDS composition analysis results of the electrodeposited films formed in artificial and natural seawater at different current density values. All the investigated deposits contain Mg, Ca, O, Na, and Cl as major elements.

Discussion:

Lines 224-226: Please correct” “..the cathodic current is applied...””.......   relatively high OH- is generated producing an increase of the pH

Line 232: please correct: “.. low cathodic current...”

Author Response

First at all, we would like to mention our appreciation for the thorough review and the constructive feedback. We thank to you for your comments. We were able to correct our paper considering your comments. Please see the attachment for revised manuscripts. 

Reviewer 4 Report

publish in the present form 

Author Response

Thank your for comments.